# Temporal characteristics of aspiration pneumonia in elderly inpatients: From resumption of oral intake to onset

**Daisuke Furukawa**[1]*, **Yoshitaka Yamanaka**[1,2], **Hajime Kasai**[3,4], **Takashi Urushibara**[3], **Tomokazu Ishiwata**[1], **Sachiyo Muranishi**[1‡]

**1** Department of Rehabilitation, Kimitsu Chuo Hospital, Kisarazu, Japan, **2** Department of Neurology, Urayasu Rehabilitation Education Center, Chiba University Hospital, Chiba, Japan, **3** Department of Respirology, Kimitsu Chuo Hospital, Kisarazu, Japan, **4** Department of Respirology, Graduate School of Medicine, Chiba University, Chiba, Japan

‡ SC also contributed equally to this work.
* daisuke.furukawa01@gmail.com

## Abstract

### Background

Elderly inpatients who develop fevers after resumption of oral intake are often considered to have aspiration pneumonia (AP) and be tentatively fasted. Fasting has been associated with prolonged hospital stays and decreased swallowing ability. The purpose of this study was to compare AP and other infections after resumption of oral intake in elderly inpatients and to identify the clinical characteristics.

### Patients and methods

The records of patients who were admitted to a public tertiary hospital and referred for evaluation of swallowing disability were retrospectively reviewed to identify those who had developed AP, non-AP, or urinary tract infection (UTI) after resumption of oral intake. Eligible patients were enrolled consecutively in the study. The patient characteristics, physical findings, laboratory data, oral intake status at the time of onset of symptoms, and rate of discontinuation of oral intake after onset of infection were compared between the three types of infection.

### Results

A total of 193 patients developed an infectious illness after resuming oral intake. Among them, 114 patients had a diagnosis of AP (n = 45), non-AP (n = 24), or UTI (n = 45). There were no significant differences in patient characteristics, physical findings or laboratory data between the group with AP and the other two groups. AP developed at a median of 6 (range 1–16) days after resumption of oral intake. The rate of discontinuation of oral intake was 91.1% in the AP group, 58.3% in the non-AP group, and 26.7% in the UTI group, respectively.

**Data Availability Statement:** All relevant data are within the paper and its Supporting Information files.

**Funding:** The authors received no specific funding for this work.

**Competing interests:** The authors have declared that no competing interests exist.

## Conclusion

Infectious diseases other than AP should be considered in the differential diagnosis when nosocomial fever develops in elderly inpatients more than 17 days after resuming oral intake. Furthermore, nosocomial fever after resuming oral intake has many causes other than AP, and discontinuation of oral intake should be carefully considered.

## Introduction

Pneumonia is an age-related condition that imposes a substantial burden on the aging population. The most common etiology for pneumonia in this age group is aspiration [1]. Teramoto et al reported that approximately 80% of patients aged 70 years or older with a diagnosis of pneumonia had the aspiration type and that the incidence of aspiration pneumonia (AP) was 86.7% in patients of this age group who developed hospital-acquired pneumonia [2].

Previous systematic literature reviews identify risk factors for AP, including old age, male gender, lung diseases, dysphagia, diabetes mellitus, poor oral health, severe dementia, Parkinson's disease, malnutrition, and the use of antipsychotic drugs, proton pump inhibitors, and angiotensin-converting enzyme inhibitors [3, 4]. From among these, dysphagia is an especially important risk factor [3, 5–7]. Care should be taken to prevent aspiration in elderly inpatients suspected to have dysphagia. A videofluoroscopic swallowing study (VFSS) or fiberoptic endoscopic evaluation of swallowing (FEES) are considered gold standards for identifying patients with dysphagia. However, due to the lack of hospital facilities and staff, it is not feasible to perform one of these studies in every inpatient who develops fever after resumption of oral intake. Hospitalized elderly patients often develop febrile infections, most commonly urinary tract infections (UTIs), followed by pneumonia [8, 9]. Inpatients who develop fever after resumption of oral intake are often considered to have AP as a result of dysphagia and may be tentatively fasted by the clinician. Although resumption of oral intake increases the risk of aspiration of food and thereby the risk of AP, fasting after admission to hospital has been associated with a prolonged hospital stay and decreased swallowing ability [10].

In clinical practice, when a patient at risk of AP resumes oral intake, it is difficult to decide the duration for which they should be monitored for signs of aspiration and whether to discontinue oral intake when fever occurs. If there is a difference, in terms of factors such as clinical findings, temporal characteristics, and oral intake status, between patients with AP and those with other infectious diseases that can develop after resumption of oral intake, this difference could be used to guide the decision whether to continue oral intake in patients with fever that develops in hospital. However, to our knowledge, no studies have compared the clinical details of these infections, focusing on the duration between the resumption of oral intake and the onset of fever. The purpose of this study was to compare AP and other infections after resumption of oral intake in elderly inpatients and to identify the clinical characteristics.

## Patients and methods

### Patients

The records of 1378 inpatients (in a 660-bed public tertiary hospital) who were referred to a rehabilitation department for evaluation of swallowing ability between June 2014 and May 2019 were retrospectively reviewed. The study inclusion criteria were age ≧65 years, fever after resuming oral intake, and development of an infectious illness requiring antimicrobial therapy

[11, 12]. The exclusion criteria were as follows: discontinuation of oral intake (because of terminal cancer, neurodegenerative disease, surgery, recurrence of a cerebrovascular disorder, disturbance of consciousness by psychotropic medication), infection other than pneumonia or UTI, and multiple sources of infection. Oral care for the prevention of AP is recommended at our hospital [13], and routine oral care was performed in the ward by a nurse whenever a patient required assistance with self-care. If a patient had poor oral hygiene, treatment was provided by a dentist and oral hygienist.

The study protocol was approved by The Clinical Research Ethics Committee of Kimitsu Chuo Hospital (approval number 454).

## Definitions

The date of onset of infection was defined as the first day when body temperature was noted to be ≧37.5˚C [14] after resuming oral intake and antimicrobial therapy was started. Resuming oral intake was defined as continuous ingestion of more than 10% of daily meals. Discontinuation of oral intake was defined as 1 day or more of fasting as determined by the clinician.

Pneumonia was defined as new infiltrates on chest radiograph and/or computed tomography and abnormal findings on laboratory data (white blood cell count ≧10,000/μL and/or increased level of serum C-reactive protein) [15, 16]. AP was defined as pneumonia that occurred in patients with dysphagia or aspiration based on a clinical swallowing evaluation. UTI was defined as a condition showing both pyuria and bacteriuria [17, 18]. The initial diagnosis of infection was made by a clinician. After that, medical records, laboratory, and radiologic findings were reviewed by the authors to confirm the diagnoses and patients who did not meet the definitions were excluded.

Swallowing function had been assessed in all patients by two speech-language-hearing therapists using dysphagia screening consisting of a repetitive saliva swallowing test [19], a modified water swallowing test [20], a food test [20], and cervical auscultation [21]. The cervical auscultation findings were compared with "clear" expiration before water was swallowed, and when breathing, sound wetness, stridor, coughing, throat clearing, or voice distortion was judged as "abnormal." Patients in whom these tests could not exclude the possibility of dysphagia underwent a more detailed swallowing evaluation in the form of FEES and/or VFSS performed by an otolaryngologist. FEES findings were described for accumulated oropharyngeal secretions according to the Murray Secretion Scale (MSS) score [22]. The MSS score ranges from 0 (no secretions) to 3 (secretions in the laryngeal vestibule). VFSS findings were described using the Penetration-Aspiration Scale (PAS) score [23] for each bolus consistency (liquid, puree, and solid). The PAS score ranges from 1 (normal) to 8 (silent aspiration), and the worst PAS score in the three different boluses was used. The MSS and PAS scores were derived from the video recordings by a speech-language-hearing therapist who was blinded to the clinical data. The Functional Oral Intake Scale (FOIS) was used to assess oral intake at discharge [24]. The FOIS score ranges from level 1 (nothing by mouth) to level 7 (full oral intake).

## Measurements

Data on age, sex, underlying disease (cerebrovascular, gastrointestinal, respiratory, circulatory, orthopedic, and other), past medical history, and comorbidities (head and neck tumors, gastroesophageal surgery, cerebrovascular disease, chronic lower respiratory airway disease, diabetes mellitus, and dementia), body mass index (BMI), frequency of use of urinary catheters, frequency of use of proton pump inhibitors and angiotensin-converting enzyme inhibitors, length of hospital stay, time from hospitalization to resuming oral intake, mortality, and

swallowing ability (positive dysphagia screening rate, MSS, PAS, and FOIS scores at discharge) were collected from the medical records or from video recordings after referencing earlier studies of factors that may affect oral intake [25–27].

The following clinical information obtained at the time of onset of infection after resuming oral intake was also collected: physical findings (Glasgow Coma Scale, Eastern Cooperative Oncology Group performance status [28], pulse rate, systolic blood pressure, and peak body temperature), laboratory data (albumin, blood urea nitrogen, C-reactive protein levels, and white blood cell count), oral intake status (a dysphagia diet, and assistance with meals), interval between resuming oral intake and onset of the infectious illness, and rate of discontinuation of oral intake. Information on oral intake status was collected from the medical records or the nursing records.

## Statistical analysis

Continuous and ordinal data are presented as the median (interquartile range or range) and differences were analyzed using the Kruskal–Wallis test. Categorical data are expressed as the number of participants (percentage) and differences were analyzed using Fisher's exact test or the chi-squared test. Bonferroni correction was used to correct for multiple comparisons. All statistical analyses were performed with EZR (Saitama Medical Center, Jichi Medical University, Saitama, Japan), which is a graphical user interface for R (The R Foundation for Statistical Computing, Vienna, Austria). More specifically, it is a modified version of R commander designed to add statistical functions frequently used in biostatistics [29]. P-values <0.05 were considered statistically significant.

## Results

A flow chart showing the patient selection process is provided in Fig 1. One hundred and ninety-three of 1378 inpatients developed an infectious illness after resuming oral intake before or after being referred to the rehabilitation department for evaluation of swallowing ability. Seventy-nine patients were excluded (discontinuation of oral intake, n = 33; infection other than pneumonia or UTI, n = 35; multiple sources of infection, n = 3; Pneumonia or AP without radiologic findings, n = 2; UTI without pyuria or bacteriuria, n = 6). After these exclusions, data for 114 patients (median age 78.0 [interquartile range, 74.0–85.0] years) were available for inclusion in the analysis. The diagnosis was AP in 45 of these patients, non-AP in 24, and UTI in 45.

There were no significant differences in patient characteristics between the group with AP and the other two groups (Table 1). The positive dysphagia screening rate, MSS, and PAS scores were significantly higher in the AP group than in the non-AP group (P<0.01, P<0.05, and P<0.01, respectively) and the UTI group (P<0.01, P<0.01 and P<0.01, respectively). The FOIS score at discharge was significantly lower in the AP group than in the non-AP group (P<0.05) and UTI group (P<0.01; Table 1). These results indicate that swallowing ability was significantly lower in the AP group than in the other two groups. The median interval between resuming oral intake and onset of the infectious illness was 6 (range, 1–16) days in the AP group, 9 (2–49) days in the non-AP group, and 9 (2–40) days in the UTI group (Fig 2). This interval was significantly shorter in the AP group than in the non-AP group (P<0.05) and UTI group (P<0.01). AP did not develop later than 17 days after resuming oral intake.

Table 2 shows the clinical findings at the time of onset of infection. There were no significant differences in the physical findings, laboratory data, or oral intake status between the group with AP and the other two groups. Fig 3 shows the rate of discontinuation of oral intake after onset of infection. The discontinuation rate was 91.1% in the AP group, 58.3% in the non-AP group, and 26.7% in the UTI group. When both the non-AP group and the UTI group

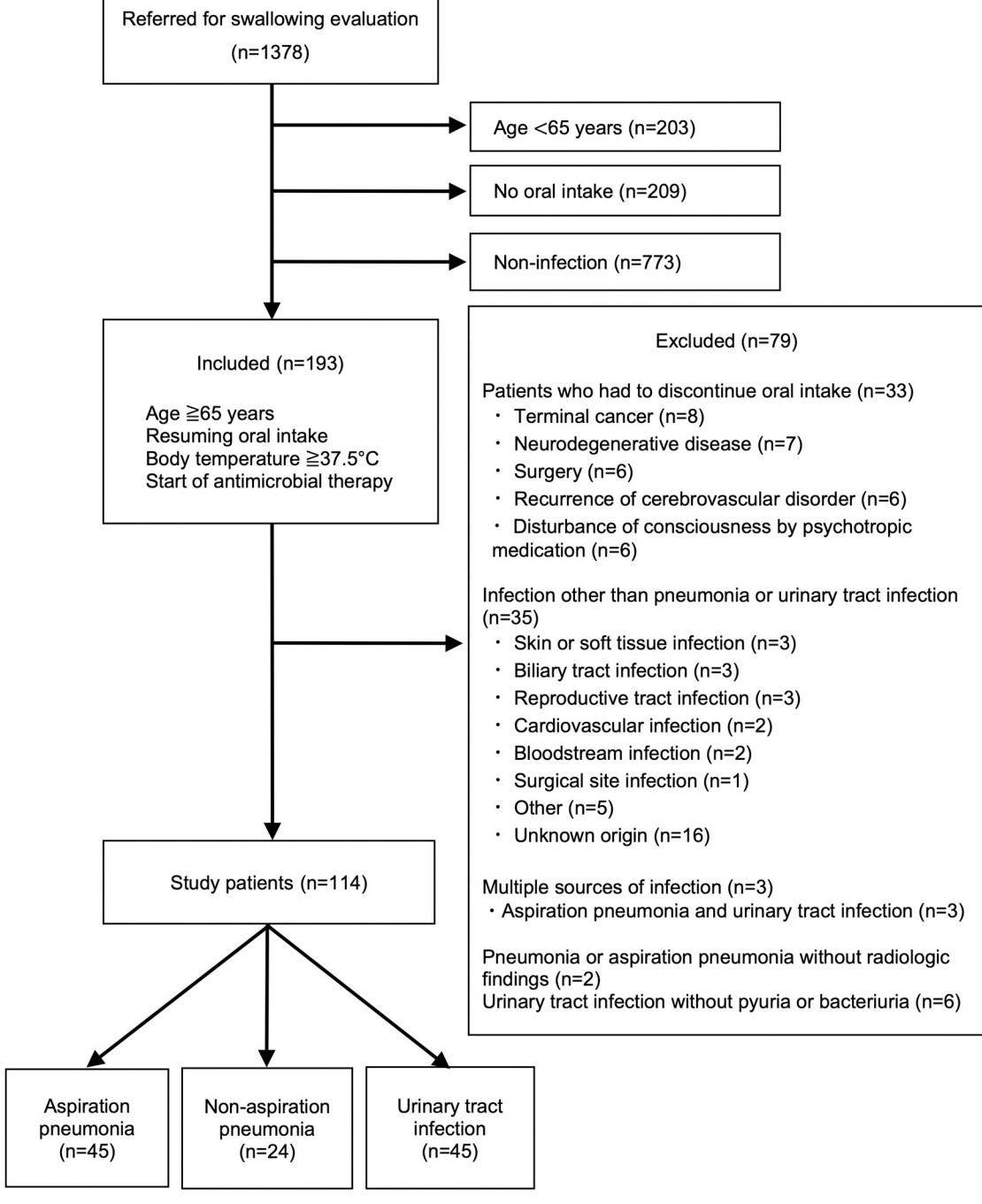

**Fig 1. Flow chart showing how patients were selected for enrolment in the study.**

developed after resuming oral intake were combined, 42.5% of patients were discontinued oral intake.

## Discussion

The results of this study focused on the clinical characteristics of elderly patients who developed AP after resuming oral intake during an hospital admission. Our study indicated two

**Table 1. Patient characteristics and swallowing function.**

| Variable | All | Aspiration pneumonia | Non-aspiration pneumonia | Urinary tract infection | P-value |
|---|---|---|---|---|---|
| | (n = 114) | (n = 45) | (n = 24) | (n = 45) | |
| Age, years, median (IQR) | 78.0 (74.0–85.0) | 79.0 (75.0–85.0) | 78.0 (74.0–81.3) | 77.0 (72.0–85.0) | 0.40 |
| Sex, male, n (%) | 79 (69.3) | 32 (71.1) | 20 (83.3) | 27 (60.0) | 0.15 |
| Underlying disease, n (%) | | | | | |
| Cerebrovascular | 42 (36.8) | 15 (33.3) | 4 (16.7)* | 23 (51.1)* | <0.05 |
| Gastrointestinal | 22 (19.3) | 11 (24.4) | 6 (25.0) | 5 (11.1) | 0.20 |
| Respiratory | 16 (14.0) | 8 (17.8) | 6 (25.0)* | 2 (4.4)* | <0.05 |
| Orthopedic | 11 (9.6) | 6 (13.3) | 1 (4.2) | 4 (8.9) | 0.51 |
| Circulatory | 9 (7.9) | 2 (4.4) | 2 (8.3) | 5 (11.1) | 0.55 |
| Other | 14 (12.3) | 3 (6.7) | 5 (20.8) | 6 (13.3) | 0.21 |
| Past medical history and comorbidities, n (%) | | | | | |
| Head and neck tumor | 1 (0.9) | 0 | 1 (4.2) | 0 | 0.21 |
| Gastroesophageal surgery | 17 (14.9) | 9 (20.0) | 5 (20.8) | 3 (6.7) | 0.13 |
| Cerebrovascular disease | 27 (23.7) | 13 (28.9) | 7 (29.2) | 7 (15.6) | 0.27 |
| Chronic lower respiratory airway disease | 7 (6.1) | 3 (6.7) | 3 (12.5) | 1 (2.2) | 0.22 |
| Diabetes mellitus | 31 (27.2) | 10 (22.2) | 8 (33.3) | 13 (28.9) | 0.59 |
| Dementia | 17 (14.9) | 6 (13.3) | 6 (26.1) | 5 (11.4) | 0.31 |
| BMI kg/m$^2$, median (IQR) | 20.9 (18.3–23.1) | 20.1 (17.9–22.4) | 19.2 (16.9–22.0)* | 22.3 (20.1–25.0)* | <0.05 |
| Missing, n (%) | 6 (5.2) | 3 (6.7) | 1 (4.2) | 2 (4.4) | |
| Use of a urinary catheter, n (%) | 52 (45.6) | 17 (37.8) | 10 (41.7) | 25 (55.6) | 0.24 |
| Use of a proton pump inhibitor, n (%) | 73 (64.0) | 31 (68.9) | 15 (62.5) | 27 (60.0) | 0.67 |
| Use of a angiotensin-converting enzyme inhibitor, n (%) | 7 (6.1) | 3 (6.7) | 0 | 4 (8.9) | 0.52 |
| Length of hospital stay, days, median (IQR) | 43 (31–60) | 47 (32–57) | 40 (35–61) | 43 (31–64) | 0.99 |
| Time from hospital admission to resuming oral intake, day, median (IQR) | 4 (2–12) | 4 (2–10) | 2 (1–12) | 5 (2–15) | 0.29 |
| Mortality, n (%) | 18 (15.8) | 11 (24.4) | 4 (16.7) | 3 (6.7) | 0.06 |
| Swallowing function | | | | | |
| Dysphagia screen positive, n (%) | | | | | |
| RSST | 89 (78.1) | 37 (82.2) | 16 (66.7) | 36 (80.0) | 0.30 |
| MWST | 49 (43.0) | 24 (53.3) | 6 (25.0) | 19 (42.2) | 0.08 |
| FT | 17 (14.9) | 11 (24.4) | 4 (16.7)* | 2 (4.4)† | <0.05 |
| CA | 45 (39.5) | 33 (73.3) | 5 (20.8)†† | 7 (15.6)†† | <0.01 |
| Total positive rate | 50.0 (44.2) | 26.3 (59.8) | 7.8 (32.5)†† | 16.0 (35.6)†† | <0.01 |
| FEES findings | n = 33 | n = 21 | n = 5 | n = 7 | |
| MSS, median (IQR) | 1 (0–3) | 3 (1–3) | 0 (0–1)† | 0 (0–1)†† | <0.01 |
| VFSS findings | n = 38 | n = 23 | n = 7 | n = 8 | |
| PAS, median (IQR) | 7 (1–8) | 8 (7–8) | 1 (1–1)†† | 1 (1–1)†† | <0.01 |
| FOIS at discharge, median (IQR) | 4 (1–5) | 1 (1–4) | 5 (4–6)† | 5 (4–5)†† | <0.01 |

Missing, BMI not obtained in five patients; BMI, body mass index; CA, cervical auscultation; FEES, fiberoptic endoscopic evaluation of swallowing; FOIS, functional oral intake scale; FT, food test; IQR, interquartile range; MSS, Murray secretion scale; MWST, modified water swallowing test; RSST, repetitive saliva swallowing test; PAS, penetration-aspiration scale; VFSS, videofluoroscopic swallowing study.

*No significant differences vs. aspiration pneumonia group by Bonferroni correction

†P<0.05 vs. aspiration pneumonia group by Bonferroni correction

††P<0.01 vs. aspiration pneumonia group by Bonferroni correction.

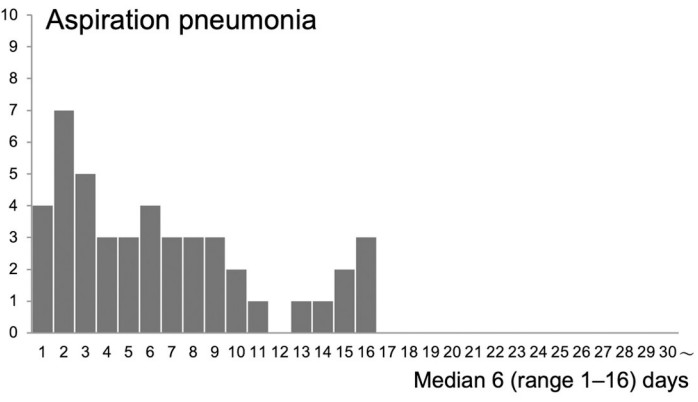

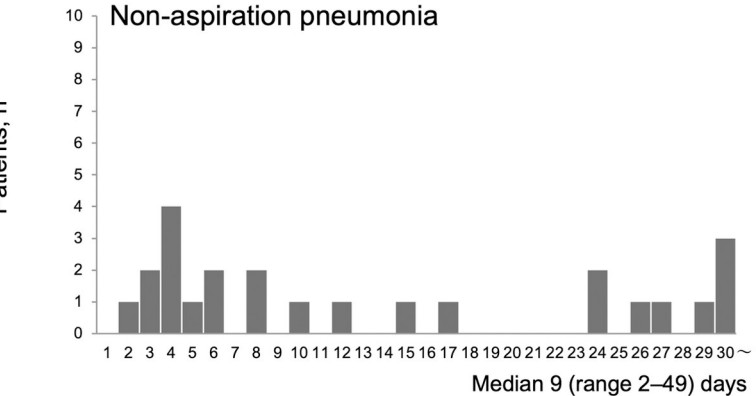

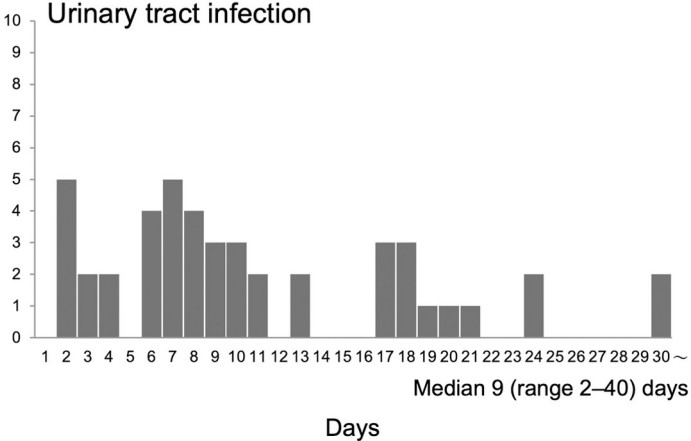

**Fig 2. Interval between resuming oral intake and onset of infection.** This interval was significantly shorter in the group with aspiration pneumonia than in the group with non-aspiration pneumonia (P<0.05) and the group with urinary tract infection (P<0.01). AP did not develop later than 17 days after resuming oral intake.

important findings. First, the interval between resuming oral intake and developing AP was shorter than that for other infections and AP did not develop later than 17 days after resuming oral intake. Second, when both the non-AP group and the UTI group developed after resuming oral intake were combined, 42.5% of patients were discontinued oral intake.

In our study, AP did not develop later than 17 days after resuming oral intake. To our knowledge, this is the first study to focus on the interval between resuming oral intake and the

**Table 2. Clinical findings and laboratory data at the onset of infection.**

| Variable | All | Aspiration pneumonia | Non-aspiration pneumonia | Urinary tract infection | P-value |
|---|---|---|---|---|---|
|  | (n = 114) | (n = 45) | (n = 24) | (n = 45) |  |
| Physical findings, median (IQR) |  |  |  |  |  |
| GCS | 14 (14–15) | 14 (14–15) | 15 (14–15) | 14 (14–15) | 0.37 |
| ECOG PS | 4 (4–4) | 4 (4–4) | 4 (3–4) | 4 (4–4) | 0.64 |
| Pulse rate (beats/min) | 86 (76–102) | 85 (74–105) | 88 (74–102) | 85 (78–98) | 0.93 |
| Systolic blood pressure (mmHg) | 121 (109–138) | 121 (109–143) | 125 (105–133) | 120 (110–138) | 0.81 |
| Peak body temperature (°C) | 38.3 (38.0–38.8) | 38.3 (38.1–38.6) | 38.3 (37.8–38.8) | 38.2 (38.1–39.1) | 0.74 |
| Laboratory data, median (IQR) |  |  |  |  |  |
| Albumin (g/dL) | 2.7 (2.3–3.1) | 2.7 (2.3–3.1) | 2.5 (2.1–2.9)* | 2.8 (2.4–3.2)* | <0.05 |
| BUN (mg/dL) | 18.1 (11.1–29.7) | 19.2 (13.0–33.7) | 22.1 (13.0–31.8) | 14.7 (10.2–23.5) | 0.07 |
| CRP (mg/dL) | 7.0 (3.9–12.4) | 8.8 (3.7–13.7) | 6.9 (5.3–8.9) | 6.7 (3.0–8.6) | 0.45 |
| WBC (×10³/μL) | 11.6 (8.6–14.7) | 12.4 (9.7–14.3) | 10.6 (8.4–13.9) | 10.8 (8.0–15.0) | 0.43 |
| Oral intake status, n (%) |  |  |  |  |  |
| Dysphagia diet | 82 (71.9) | 35 (77.8) | 15 (62.5) | 32 (71.1) | 0.44 |
| Assistance with meals | 49 (43.0) | 16 (35.6) | 13 (54.2) | 20 (44.4) | 0.32 |

BUN, blood urea nitrogen; CRP, C-reactive protein; ECOG PS, Eastern Cooperative Oncology Group performance status; GCS, Glasgow Coma Scale; IQR, interquartile range; WBC, white blood cell count.

*No significant differences vs. aspiration pneumonia group by Bonferroni correction;†P<0.05 vs. aspiration pneumonia group by Bonferroni correction; ††P<0.01 vs. aspiration pneumonia group by Bonferroni correction.

onset of AP. Mandell, et al. described that AP is usually acute, with symptoms developing within hours to a few days after a sentinel event, although anaerobic aspiration may be subacute because of the less virulent bacteria [30]. In our study, the interval between resuming oral intake and onset of AP ranged from 1 to 16 days and the present results support the possibility that AP develops from acute to subacute. AP is associated with dysphagia and caused by inhalation of saliva or food [31]. Although aspiration is an essential feature of AP, many episodes are unwitnessed [30] and it is difficult to gather clinically relevant information concerning the aspiration process. Therefore, it is noteworthy that this study focused on resuming oral intake and found a temporal relationship with development of AP. Our findings indicate that elderly patients admitted to an hospital should be observed more carefully for signs and symptoms of AP for 16 days after resuming oral intake.

Oral intake was withheld in about 40% of patients, including the UTI group. This result shows that fasting was implemented unnecessarily in patients with infections other than AP. Furthermore, these patients were referred to a speech-language-hearing therapist for a swallowing evaluation, indicating that nosocomial fever after resuming oral intake in the elderly tends to be suspected initially to be AP. Fasting during a hospital admission has adverse consequences for patients, including a prolonged treatment duration and a decline in swallowing ability [10]. Therefore, when a patient develops fever after oral intake, possibilities other than AP should be considered to prevent unnecessary fasting.

This study has several limitations, in particular its single-center retrospective design and small sample size. Most cases of AP develop from a multifactorial background. Pulmonary syndromes caused by aspiration differ depending on the amount and nature of the material aspirated, the frequency of aspiration, and the host response [32]. More data are needed to understand how each factor influences the time course of AP development. Second, since we included patients who were referred by a clinician for suspected dysphagia, our results may not be applicable to elderly hospitalized patients in general. Nevertheless, the results are

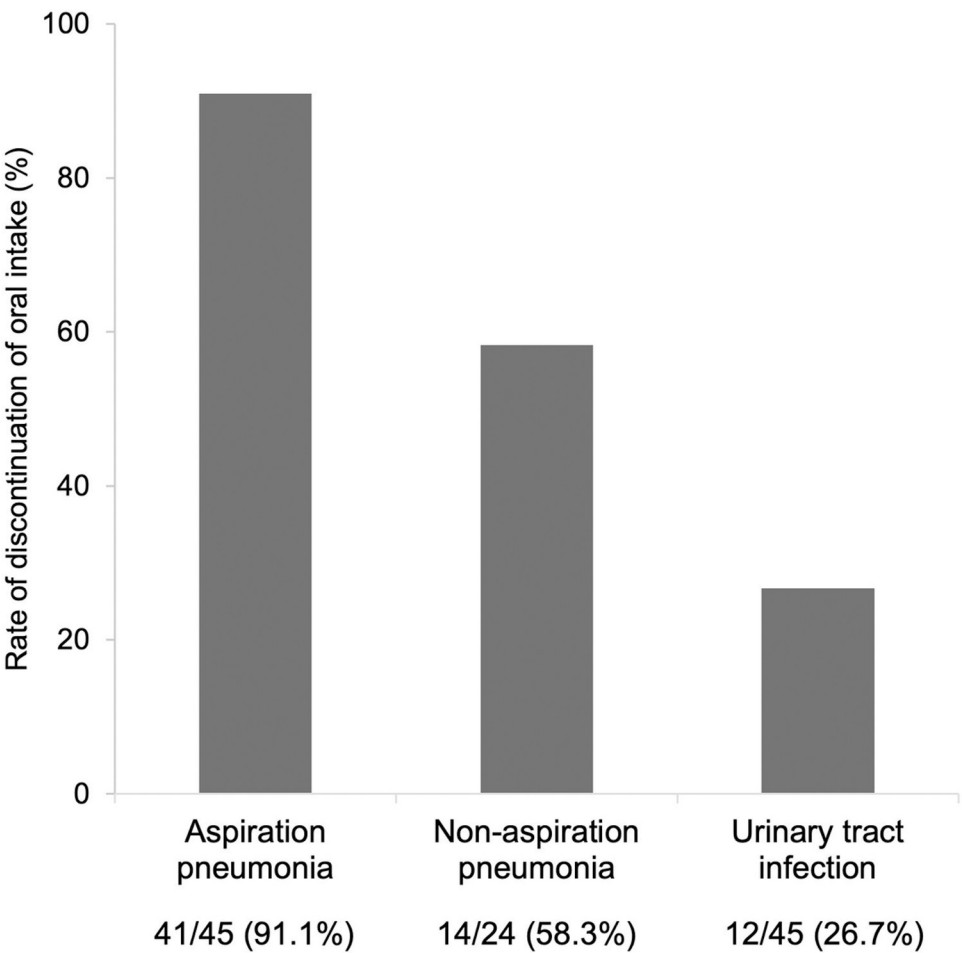

**Fig 3. Rate of discontinuation of oral intake after onset of infection.** When both the non-AP group and the UTI group developed after resuming oral intake and were combined, 42.5% of patients were discontinued oral intake.

clinically relevant for patients with a risk for AP. Third, AP was diagnosed on clinical grounds by the clinician and therefore susceptible to misclassification. However, swallowing ability was evaluated using standardized swallowing evaluations in all patients; patients who did not meet the definition were excluded. Therefore, we believe that our findings are reliable.

## Conclusion

Elderly patients admitted to an hospital should be observed more carefully for signs and symptoms of AP for 16 days after resuming oral intake. Furthermore, since fever occurs more than 17 days after resumption of oral intake, unnecessary fasting may be avoided by considering infections other than AP as a differential diagnosis. The findings of this study may help guide decisions whether or not to continue oral intake in patients with fever that develops in hospital. Prospective multicenter studies with large sample sizes are now needed to confirm our results.

## Supporting information

**S1 File. Dataset of this study.**
(XLSX)

## Acknowledgments

We would like to thank Editage (www.editage.jp) for English language editing.

## Author Contributions

**Conceptualization:** Daisuke Furukawa, Yoshitaka Yamanaka, Hajime Kasai, Takashi Urushibara, Tomokazu Ishiwata, Sachiyo Muranishi.

**Data curation:** Daisuke Furukawa, Tomokazu Ishiwata, Sachiyo Muranishi.

**Formal analysis:** Daisuke Furukawa, Tomokazu Ishiwata, Sachiyo Muranishi.

**Funding acquisition:** Daisuke Furukawa.

**Investigation:** Daisuke Furukawa, Tomokazu Ishiwata, Sachiyo Muranishi.

**Methodology:** Daisuke Furukawa, Yoshitaka Yamanaka, Hajime Kasai, Takashi Urushibara, Tomokazu Ishiwata, Sachiyo Muranishi.

**Project administration:** Daisuke Furukawa, Sachiyo Muranishi.

**Resources:** Daisuke Furukawa, Tomokazu Ishiwata, Sachiyo Muranishi.

**Software:** Daisuke Furukawa.

**Supervision:** Daisuke Furukawa, Yoshitaka Yamanaka, Hajime Kasai, Takashi Urushibara, Sachiyo Muranishi.

**Validation:** Daisuke Furukawa.

**Visualization:** Daisuke Furukawa.

**Writing – original draft:** Daisuke Furukawa, Sachiyo Muranishi.

**Writing – review & editing:** Daisuke Furukawa, Yoshitaka Yamanaka, Hajime Kasai, Takashi Urushibara, Tomokazu Ishiwata, Sachiyo Muranishi.

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
