## [Decision Letter · Decision Letter 0]

18 Oct 2021

PONE-D-21-20197Temporal  characteristics of aspiration pneumonia in elderly inpatients: From resumption of oral intake to onsetPLOS ONE

Dear Dr. Furukawa,

Thank you for submitting your manuscript to PLOS ONE. After careful consideration, we feel that it has merit but does not fully meet PLOS ONE’s publication criteria as it currently stands. Therefore, we invite you to submit a revised version of the manuscript that addresses the points raised during the review process.

ACADEMIC EDITOR: Please review comments made by the reviewers and provide point by point response in your revised manuscript.

We look forward to receiving your revised manuscript.

Kind regards,

Muhammad Adrish, MD, MBA, FCCP, FCCM

Academic Editor

PLOS ONE

Journal Requirements:

Reviewers' comments:

Reviewer's Responses to Questions

**Comments to the Author**

1. Is the manuscript technically sound, and do the data support the conclusions?

Reviewer #1: Yes

Reviewer #2: Partly

2. Has the statistical analysis been performed appropriately and rigorously? 

Reviewer #1: Yes

Reviewer #2: Yes

3. Have the authors made all data underlying the findings in their manuscript fully available?

Reviewer #1: Yes

Reviewer #2: Yes

4. Is the manuscript presented in an intelligible fashion and written in standard English?

Reviewer #1: Yes

Reviewer #2: Yes

5. Review Comments to the Author

Reviewer #1: Furukawa et al in an interesting study analyze the temporal characteristics of aspiration pneumonia (AP) in the elderly inpatients after resumption of oral intake. The authors conclude that close observation for AP should be kept for 16 days after resuming oral intake. Fever observed after that oral eating resuming date might be due to non-AP or to an urinary tract infection (UTI). Obviously, in these febrile patients without AP, discontinuation of oral intake is useless , extended hospitalization stay and might be even harmful. The work presented here is certainly original in my opinion and brings forward the relevant issue of the timing of AP in hospitalized elderly patients that restart oral intake. This reviewer adds some minor points that perhaps might clarify some aspects of the manuscript:

1.How angiotensin-converting enzyme (ACE) inhibitors are associated with aspiration pneumonia (AP)?Could you discuss it briefly?(page 3, line 50)

2.What is the purpose of cervical auscultation ? Looking for stridor due to upper airways stenosis?Looking for pharingo-esophageal bubbling after oral intake? (page 5, line 104)

3.Fig.2 key description that AP do not develop later than 17 days after resuming oral intake lacks in the Results section although it appears in the Discussion Section as well as in the absctract. I would added it also to the Results section due to its importance. (page 11, line 187)

Reviewer #2: Although the authors make an important finding of developing AP symptoms after resuming food intake has a smaller interval, the data is insufficient to support the conclusions. The result section is short and confusing. The authors make a point about how resumption of food intake could be leading to AP. However, they also say that onset of AP symptoms leads to the discontinuation of food intake in figure 3, which is not unusual. Hence, the purpose of figure 3 is unclear and difficult to interpret.

6. PLOS authors have the option to publish the peer review history of their article (what does this mean?). If published, this will include your full peer review and any attached files.

Reviewer #1: **Yes: **VICTOR ASENSI

Reviewer #2: No

---

## [Author Response · Author response to Decision Letter 0]

21 Dec 2021

Reviewer #1: 

Furukawa et al in an interesting study analyze the temporal characteristics of aspiration pneumonia (AP) in the elderly inpatients after resumption of oral intake. The authors conclude that close observation for AP should be kept for 16 days after resuming oral intake. Fever observed after that oral eating resuming date might be due to non-AP or to an urinary tract infection (UTI). Obviously, in these febrile patients without AP, discontinuation of oral intake is useless, extended hospitalization stay and might be even harmful. The work presented here is certainly original in my opinion and brings forward the relevant issue of the timing of AP in hospitalized elderly patients that restart oral intake. This reviewer adds some minor points that perhaps might clarify some aspects of the manuscript:

Thank you for all of your detailed comments and suggestions. We found them quite useful as we approached our revision. We are grateful for the time and energy you expended on our behalf. 

Comment 1: How angiotensin-converting enzyme (ACE) inhibitors are associated with aspiration pneumonia (AP)? Could you discuss it briefly? (page 3, line 50)

Response:

We agree with the reviewer's advice. ACE inhibitors are listed as a risk factor for AP but were not included in the analysis of this study. Therefore, we added the use of ACE inhibitors to the analysis. The result showed that the use of ACE inhibitors was very low (7 out of 114 patients) and not significantly different among the three infections. Thus, we have only described the results and not discussed the association.

Proton pump inhibitors were also a point we had missed; we have now added them to our analysis. The results and changes are in Table 1, Measurements section (p.6, line 126) and Supporting information “S1 File”.

Comment 2: What is the purpose of cervical auscultation? Looking for stridor due to upper airways stenosis? Looking for pharingo-esophageal bubbling after oral intake? (page 5, line 104)

Response:

Thanks for this suggestion. Per your comment, we have added the assessment procedure of cervical auscultation to the Definitions section (p.5, lines 105-108): “The cervical auscultation findings were compared with “clear” expiration before water was swallowed, and when breathing, sound wetness, stridor, coughing, throat clearing, or voice distortion was judged as “abnormal.””

Comment 3: Fig.2 key description that AP do not develop later than 17 days after resuming oral intake lacks in the Results section although it appears in the Discussion Section as well as in the absctract. I would added it also to the Results section due to its importance. (page 11, line 187)

Response:

Thanks for pointing this out. The following sentence was added to the Results section (p.7, lines 172-173) and figure 2 legend (p.9, lines 179-180): “AP did not develop later than 17 days after resuming oral intake.”

Reviewer #2: 

Comment 1: Although the authors make an important finding of developing AP symptoms after resuming food intake has a smaller interval, the data is insufficient to support the conclusions. The result section is short and confusing. The authors make a point about how resumption of food intake could be leading to AP. However, they also say that onset of AP symptoms leads to the discontinuation of food intake in figure 3, which is not unusual. Hence, the purpose of figure 3 is unclear and difficult to interpret.

Response:

In Figure 3, we emphasized that AP was significantly more fasted than other infections; however, it was irrelevant to the discussion of this study. Thank you for identifying this inconsistency. We want to show that patients with infections other than AP were also unnecessarily fasting at fever after oral intake. Therefore, we have removed the description of significant differences from Figure 3 and the Results section. Furthermore, to emphasize the high discontinuation rate for infections other than AP, we have revised the following sentence in the Results section (p9, lines 186-187), the legend of Figure 3 (p.10, lines 191-192) and Discussion section (p,11, lines 199-200): “When both the non-AP and the UTI groups developed after resuming oral intake were combined, 42.5% of patients discontinued oral intake.” 

In addition, “about one quarter of patients" was changed to "about 40% of patients" in the Discussion section (p.11, line 215).

---

## [Decision Letter · Decision Letter 1]

4 Apr 2022

Temporal  characteristics of aspiration pneumonia in elderly inpatients: From resumption of oral intake to onset

PONE-D-21-20197R1

Dear Dr. Furukawa,

We’re pleased to inform you that your manuscript has been judged scientifically suitable for publication and will be formally accepted for publication once it meets all outstanding technical requirements.

Kind regards,

Shane Patman, PhD

Academic Editor

PLOS ONE

Additional Editor Comments (optional):

Reviewers' comments:

Reviewer's Responses to Questions

**Comments to the Author**

1. If the authors have adequately addressed your comments raised in a previous round of review and you feel that this manuscript is now acceptable for publication, you may indicate that here to bypass the “Comments to the Author” section, enter your conflict of interest statement in the “Confidential to Editor” section, and submit your "Accept" recommendation.

Reviewer #1: All comments have been addressed

Reviewer #2: All comments have been addressed

2. Is the manuscript technically sound, and do the data support the conclusions?

Reviewer #1: Yes

Reviewer #2: Partly

3. Has the statistical analysis been performed appropriately and rigorously? 

Reviewer #1: Yes

Reviewer #2: Yes

4. Have the authors made all data underlying the findings in their manuscript fully available?

Reviewer #1: Yes

Reviewer #2: Yes

5. Is the manuscript presented in an intelligible fashion and written in standard English?

Reviewer #1: Yes

Reviewer #2: Yes

6. Review Comments to the Author

Reviewer #1: I am pleased with the changes introduced in the revised version of the manuscript . The authors have adressed all the queries raised by this reviewer

Reviewer #2: (No Response)

7. PLOS authors have the option to publish the peer review history of their article (what does this mean?). If published, this will include your full peer review and any attached files.

Reviewer #1: **Yes: **VICTOR ASENSI

Reviewer #2: No

---

## [Editor Report · Acceptance letter]

6 Apr 2022

PONE-D-21-20197R1 

Temporal characteristics of aspiration pneumonia in elderly inpatients: From resumption of oral intake to onset 

Dear Dr. Furukawa:

I'm pleased to inform you that your manuscript has been deemed suitable for publication in PLOS ONE. Congratulations! Your manuscript is now with our production department. 

Kind regards, 

on behalf of

Assoc Prof Shane Patman 

Academic Editor

PLOS ONE